# Measuring the Cultivated Land Use Efficiency in China: A Super Efficiency MinDS Model Approach

**DOI:** 10.3390/ijerph20010583

**Published:** 2022-12-29

**Authors:** Guijie Qiu, Xiaonan Xing, Guanqiao Cong, Xinyu Yang

**Affiliations:** College of Biological and Agricultural Engineering, Jilin University, 5988 Renmin Street, Changchun 130022, China

**Keywords:** cultivated land use efficiency, input–output analysis, the super efficiency MinDS model, convergence test, carbon emissions

## Abstract

Cultivated land is a vital strategic resource to ensure human survival and development. It is pertinent to introduce the environmental constraint index into the study of cultivated land use efficiency and promote the coordinated development of environmental and economic benefits. Based on the construction of the evaluation index system of cultivated land use efficiency, including carbon emission as the undesired output, this paper used the super efficiency MinDS model to measure the cultivated land use efficiency of China by using the data from 2009 to 2019. The results show the following. (1) During 2009–2019, the overall use efficiency of cultivated land in China showed a tendency to fluctuate and increase, ranging from 0.871 to 0.948, indicating high use efficiency. Eleven provinces had an average land use efficiency of more than 1. (2) Among the input–output indicators, the carbon emission indicator showed the largest average percentage of improvement at 15.21%, followed by the pesticide index and agricultural machinery index, and the smallest average improvement was the irrigation indicator at 3.55%. (3) There were apparent differences in the average relaxation and improvement proportion of input–output indicators of cultivated land use in the provinces. (4) China’s cultivated land use efficiency has absolute σ-convergence, absolute β-convergence and conditional β-convergence, which means that the difference in cultivated land use efficiency between provinces has a tendency to decrease, and that China’s cultivated land use efficiency will stabilize. This paper provides a clear direction for the promotion and improvement of cultivated land use efficiency in China.

## 1. Introduction

Cultivated land guarantees global food security and is the material basis for human survival and development. However, with the rapid development of industrialization and urbanization, many areas of cultivated land have been used for other purposes besides agriculture in many countries, including China [1]. For example, the amount of cultivated land per capita in the world declined from 0.41 hectares in 1960 to 0.21 hectares in 2019 [2]. The decreasing cultivated land and the vast population base have brought significant challenges to world agricultural production [3]. In addition, to increase the output of agricultural products, the input of pesticides and fertilizers has been unreasonably increased in the process of cultivated land utilization, which has led to an increase in carbon emissions, which has led in turn to global warming, sea level rises and extreme weather [4,5,6]. Keith Paustian and Vernon Cole C. (1998) [7] believe that the carbon emissions caused by agricultural activities, which take agricultural land use as a carrier, account for approximately 20% of the total carbon emissions from human activities. Using cultivated land resources efficiently and reasonably has become the focus of sustainable development worldwide [8,9,10]. Thus, it is urgently necessary to introduce the concepts of green development and environmental constraints into the study of cultivated land use efficiency and explore methods to improve the cultivated land use efficiency [11].

Efficiency is a widely used concept in economics. Stochastic frontier methods and data envelopment analysis (DEA) are the most commonly used research methods for measuring efficiency. For example, Binam et al. (2004) [12] used a stochastic frontier model to analyze data from 450 farmer surveys in Cameroon to measure the technical efficiency of smallholder production. In addition, Wang et al. (2014) [13] employed a stochastic frontier production function in their analysis. Yang et al. (2017) [14] also measured the efficiency level of grain production in China’s North China Plain using the stochastic frontier method by selecting grain production data for four years from 2000 to 2015. Deng et al. (2019) [15] analyzed the land production efficiency in Shandong province, which was based on the stochastic frontier analysis method. Overall, the drawbacks of the stochastic frontier method are that it is difficult to determine a specific production function form and it is prone to multicollinearity problems, as its use is based on the determination of a particular function of production form [15,16]. By contrast, data envelopment analysis does not require the use of a specific form of production function, which can prevent errors when building the model. The decision-making unit (DMU) in the traditional DEA model uses the method of equal ratio (an equiproportional) improvement to measure cultivated land use efficiency, also known as the radial distance model [17,18]. As the radial distance model ignores the slack variable in the process of measuring the cultivated land use efficiency [19], scholars use the non-radial slack-based measure (SBM) model to measure the efficiency of cultivated land use, which is measured by the average proportion of improvement in each input–output index [20,21]. However, the SBM model takes the point on the strong efficient frontier that is furthest from the evaluated decision unit as the projection point, resulting in inefficiency maximization. As a result, to compensate for the shortcomings of the SBM model, the minimum distance to strong efficient frontier (MinDS) model uses the closest point to the evaluated decision unit as the projection point, which is a more realistic measure of cultivated land use efficiency [22]. To further identify the differences between the efficient decision units, Andersen (1993) proposed the super efficiency model, which measures the efficiency of the efficient decision unit by forming a new production frontier with other efficient decision units [23].

Scholars have extensively researched cultivated land use efficiency evaluation index systems [24]. For example, Kendall (1939) [25] was the first to propose a method for measuring the utilization efficiency of cultivated land based on output per unit area. Later, Ramanaiah et al. (1985) [26] used this method to assess the efficiency of cultivated land use in Andhra Pradesh. However, the single-factor index is too simple to measure the efficiency of cultivated land use. The cultivated land use must consider multiple inputs and outputs, so a multi-factor index was introduced [27]. The input index of cultivated land use efficiency is chosen primarily from the perspectives of land, capital and labor. In contrast, the output index is the grain output and agricultural gross output value [28,29,30,31]. Alternatively, with the emergence of environmental issues, cultivated land produces expected outputs, such as grain, and generates significant pollution. Other scholars have included pollution in the measurement system of cultivated land use efficiency as an indicator of undesirable output. Carbon emissions are the primary source of pollution, and the unwanted output of cultivated land utilization is generated during the fertilizer, pesticide, agricultural film, machinery and irrigation utilization processes [32,33]. For example, Liang (2012) [34] chose the agricultural land pollution emission index as the undesirable output index to measure agricultural land pollution emission in various regions. When studying the utilization efficiency of cultivated land in prominent grain-producing areas, Kuang et al. (2018) [35] also introduced carbon emission as an undesired output. Furthermore, Feng et al. (2015) [36] examined the utilization efficiency of cultivated land in China using non-point source pollution and carbon emissions as indicators of undesirable output.

In summary, although significant progress has been made in the research on cultivated land use efficiency, there are also some deficiencies. First, traditional studies of the efficiency of cultivated land use focus on improving the economic benefits, while ignoring the negative environmental impact of cultivated land use. There is a lack of literature focusing on the negative outcomes of cultivated land use. Second, many works in the literature use the cultivated land area as an input indicator to measure cultivated land use efficiency, which confuses the concepts of agricultural productivity and cultivated land use efficiency [22]. Third, most literature uses the SBM model for analysis when selecting the DEA method to measure the efficiency of cultivated land use. Although the model takes into account the relaxation and improvement of each input–output index, it takes the effective frontier projection point farthest from the decision-making unit as a reference, which amplifies the inefficiency of cultivated land use production activities.

Based on the existing research results, this paper introduces the concept of green development into the study of cultivated land use efficiency against the backdrop of environmental deterioration [37,38,39]. In terms of indicators, carbon emission, an undesirable output index, is added. Moreover, an index system is built with rice as the primary carbon source, considering that rice is a significant source of carbon emission, accounting for 42.28% of China’s total agricultural carbon emissions in agricultural production [33]. In terms of research methods, in this paper, the super efficiency model proposed by Anderson and Petersen and the minimum distance to strong efficiency frontier model (MinDS) are combined to measure cultivated land use efficiency. The combination not only overcomes the defects of the radial DEA model and the non-radial SBM model, but also further distinguishes the efficiency of effective DMUs. Furthermore, the degree of improvement when the cultivated land use efficiency reaches a sufficient level is obtained, which provides a clear direction for the promotion and improvement of cultivated land use efficiency in China. To summarize, this research aims to build an index system of the green utilization efficiency of cultivated land for reference, use the super efficiency MinDS model to measure the cultivated land use efficiency in China as a whole and in provincial regions and then provide theoretical support for the release and implementation of relevant government policies in China. 

## 2. Materials and Methods

### 2.1. Selection of Indicators

In this paper, the utilization efficiency of cultivated land is measured in China as a whole and in provincial administrative regions (excluding Hong Kong, Macao and Taiwan) according to the input–output rate in the process of cultivated land utilization [40]. Considering the impact of the economic, social and ecological environment involved in the process of cultivated land utilization, and referring to the research results of Xie et al. [38], Qi et al. [41], Lu et al. [42] and Xie et al. [43], the present research uses labor, pesticides, fertilizer, agricultural machinery power, irrigation and agricultural plastic film as input indicators. The planting output value is taken as the desirable output index, and carbon emission is taken as the undesirable output index. The specific connotations and calculation methods of each index are shown in Table 1.

As shown in Table 1, carbon emission is calculated by using the carbon emission coefficient method. Since rice, as a significant crop, produces a large amount of carbon emission in the process of cultivated land production, it is added as one of the central carbon sources in cultivated land utilization. Moreover, in line with the literature [33,43,44,45,46], seven major carbon sources, including pesticides, fertilizer, agricultural film, agricultural machinery, irrigation, tillage and rice, are selected for calculation. Total carbon emissions can be obtained from Formula (1).
(1)$$E=\sum_{i=1}^{7}E_{i}=\sum_{i=1}^{7}T_{i}\delta_{i}$$


In Formula (1), *E* is the total carbon emissions; $E_{i}$ is each carbon source’s carbon emission; $T_{i}$ is each carbon source’s quantity; $\delta_{i}$ is the carbon emission coefficient of each carbon source; the sown area represents the plowing area; the carbon emission of rice is calculated by the planting area; the growth cycle of rice is calculated by the median value of 130 days. The carbon emission coefficient of each carbon source is shown in Table 2.

Due to the improved statistical technology available for the calculation of the cultivated land area, compared with 2008 [52], the cultivated land area collected data after 2009 have been dramatically improved [53], significantly impacting the input–output index of unit cultivated land area. Therefore, to ensure data comparability, this paper uses the data from 2009 to 2019 to calculate the cultivated land use efficiency.

### 2.2. Data Source

The data in this paper come from the China Statistical Yearbook, the China Rural Statistical Yearbook, the China Agricultural Machinery Industry Yearbook and the Provincial Statistical Yearbooks. It should be noted that the indicator of total power of agricultural machinery in China’s Agricultural Machinery Industry Yearbook does not include the power of three-wheeled vehicles and low-speed trucks since 2016, so the power of three-wheeled vehicles and low-speed trucks has been removed from the total power of agricultural machinery from 2009 to 2015 to maintain consistency among the statistical indicators.

### 2.3. Methods

In this study, the super efficiency model and the MinDS model are combined to measure the cultivated land use efficiency. The MinDS model is a type of non-radial distance model in DEA. It solves the problem of relaxing variables in the radial distance model and avoids the problem of underestimating the efficiency of the decision-making unit (DMU) in the SBM model. On the other hand, the MinDS model selects the nearest point to the evaluated decision-making unit on the strong, efficient production front as the projection point, so we can evaluate the improvement degree of each input–output index more accurately by applying this model. 

The model assumes that there are *n* DMUs, denoted as *DMU_j_* (*j* = 1, 2, …, *n*), there are *m* inputs in each DMU, which are denoted as *x_i_* (*i* = 1, 2, …, *m*), and the output of *q* species is denoted *y_r_* (*r* = 1, 2, …, *q*). The MinDS model is shown in Equation (2) [51].
(2)$$\max\rho_{k}=\frac{\frac{1}{m}\sum_{i=1}^{m}\left(1-s_{i}^{-}/x_{ik}\right)}{\frac{1}{q}\sum_{i=1}^{q}\left(1+s_{r}^{+}/y_{rk}\right)}\;s.t.\sum_{j\in E}x_{ij}\lambda_{j}+s_{i}^{-}=x_{ik},i=1,2,\cdots ,m\;\sum_{j\in E}y_{rj}\lambda_{j}-s_{r}^{+}=y_{rk},r=1,2,\cdots ,q\;s_{i}^{-}\geq 0,i=1,2,\cdots ,m\;s_{r}^{+}\geq 0,r=1,2,\cdots ,q\;\lambda_{j}\geq 0,j\in E\;-\sum_{i=1}^{m}v_{i}x_{ij}+\sum_{r=1}^{q}\mu_{r}y_{rj}+d_{j}=0,j\in E\;v_{i}\geq 1,i=1,2,\cdots ,m\;\mu_{r}\geq 1,r=1,2,\cdots ,q\;d_{j}\leq Mb_{j},j\in E\;\lambda_{j}\leq M(1-b_{j}),j\in E\;b_{j}\in \{0,1\},j\in E\;d_{j}\geq 0,j\in E$$

In Formula (2), constraints 6 to 12 are the added mixed integer linear constraints, which are used to limit the reference benchmark to the same hyperplane; *ρ* is the efficiency value of the evaluated DMU; *s^−^* is the relaxation variable of input; *s^+^* is the relaxation variable of output; *V* and *μ* denote the weights of input and output, respectively; *λ* represents the linear combination coefficient; *E* represents the set of valid DMUs, and *M* represents a sufficiently large positive number. When *b_j_* = 0, Equation (3) and $\lambda_{j}⩽M\;$ are satisfied, and *DMU_j_* is the reference benchmark of the currently evaluated DMU.
(3)$$-\overset{m}{\underset{i=1}{\sum }}v_{i}x_{ij}+\overset{q}{\underset{r=1}{\sum }}\mu_{r}y_{ij}=0,j\in E$$


When the input–output relaxation variable is equal to 0, the efficiency of the evaluated DMU is the highest, which is an effective DMU. From the perspective of the change in returns to scale, in the process of cultivated land use, the returns to scale of production technology are constantly changing, so this paper uses the MinDS model with variable returns to scale—that is, adding constraints and free variables $\mu_{0}$ based on Equation (2). The constraint is shown in Equation (4).
(4)$$\sum_{j\in E}\lambda_{j}=1$$


At the same time, this paper quotes the super efficiency model to distinguish the efficiency differences among effective DMUs further. The model calculates the efficiency of the evaluated DMU_k_ by eliminating the evaluated DMU_k_ and referring to the frontiers of other DMUs. The model is shown in Equation (5).
(5)$$\begin{array}{c} min\rho_{SE}=\frac{1+\frac{1}{m}\sum_{i=1}^{m}s_{i}^{-}/x_{ik}}{1-\frac{1}{s}\sum_{r=1}^{s}s_{r}^{+}/y_{rk}}\\ s.t.\sum_{j=1,j\neq k}^{n}x_{ij}\lambda_{j}-s_{i}^{-}⩽x_{ik}\\ \;\;\;\;\sum_{j=1,j\neq k}^{n}y_{rj}\lambda_{j}+s_{r}^{+}⩾y_{rk}\\ \;\;\;\;\lambda ,s^{-},s^{+}⩾0\\ \;\;\;\;i=1,2,\cdots ,m;r=1,2,\cdots ,q;j=1,2,\cdots ,n\left(j\neq k\right) \end{array}$$


## 3. Results

### 3.1. Characteristic of Cultivated Land Use Efficiency in China

The data of this study were processed with the MaxDEA Ultra 9.1 software. From 2009 to 2019, the average utilization efficiency of cultivated land in China ranged from 0.871 to 0.948, of which the highest efficiency value, obtained in 2019, was 0.948, which increased by 0.028 compared with the efficiency value in 2009, showing a slow upward trend in fluctuation. The average utilization efficiency of cultivated land in China is shown in Figure 1.

### 3.2. Cultivated Land Use Efficiency by Province

Applying the super efficiency MinDS model, this study obtained the average cultivated land use efficiency of each province from 2009 to 2019. There are 11 provinces whose average cultivated land use efficiency exceeds 1—that is, the average cultivated land use efficiency of these 11 provinces is located at the front of production, and the average cultivated land use efficiency is effective. The resource utilization rate is higher and the environmental pollution level is lower in the 11 provinces than in other provinces, as shown in Figure 2.

### 3.3. Improvement Potential of Cultivated Land Use Efficiency in China

By using the super efficiency MinDS model, we can obtain the cultivated land use efficiency of China as a whole and in provincial regions, as well as the degree of improvement when the cultivated land use efficiency reaches effectiveness—in other words, the degree of reduction in input indicators and non-expected output indicators, and the degree to which the expected output indicators can be increased. From 2009 to 2019, China’s three items with the highest average relaxation improvement ratio in terms of cultivated land use efficiency were carbon emissions, pesticides and agricultural machinery. They all had improvement rates of over 10%. Furthermore, among the input indexes, the proportion of irrigation input that can be reduced is the lowest. When the output value is practical as a positive output, the improvement ratio is small, which is 3.76%. The details can be seen in Table 3.

This paper takes the 20 provinces whose cultivated land use efficiency has not reached the production frontier as the target and observes the improvement ratio of input–output indicators in the green utilization of cultivated land. Detailed information on the average relaxation and improvement ratio of input–output indicators of cultivated land use in each province can be seen in Table 4.

### 3.4. Convergence Test of Cultivated Land Use Efficiency 

This study uses the absolute σ-convergence test model, the absolute β-convergence test model and the conditional β-convergence test model to test the convergence of cultivated land use efficiency in China. 

#### 3.4.1. Absolute σ-Convergence Test

The absolute σ-convergence test is the horizontal trend of the difference in cultivated land use efficiency between different provinces. This study uses standard deviation to reflect China’s absolute convergence of cultivated land use efficiency, as shown in Formula (6).
(6)$$S=\sqrt{\left[\sum_{i}^{n}{\left(X_{i}-\bar{X}\right)}^{2}\right]/N}$$

In Formula (6), *S* is the standard deviation; *X_i_* represents the cultivated land use efficiency of each province; *N* is the number of provinces in China, which is 31 in the study; $\bar{X}$ represents the average use efficiency of cultivated land in China.

During 2009–2019, the national cultivated land use efficiency decreased from 0.2406 in 2009 to 0.1918 in 2019, as shown in Figure 3, which showed absolute σ-convergence, and the discrepancies in cultivated land use efficiency between different provinces tended to decline.

#### 3.4.2. Absolute β-Convergence Test

The absolute β-convergence test compares the initial growth rate of cultivated land use efficiency between the lowest and highest provinces, which maintains steady cultivated land use efficiency in each province. If a province with low initial cultivated land use efficiency has a faster efficiency growth rate, it has absolute β-convergence characteristics. Otherwise, the cultivated land use efficiency in the study area will stabilize over time, as shown in Formula (7).
(7)$$\ln\left(Y_{it}/Y_{i0}\right)/T=\alpha +\beta \ln\left(Y_{i0}\right)+\varepsilon_{it}$$

In Formula (7), α is a constant term; *β* is a regression coefficient; *T* represents the time span; $Y_{i0}$ represents the cultivated land use efficiency of *i* province in the initial year; $Y_{it}$ represents the cultivated land use efficiency of *i* province in year *t*; *ε* is the random error term.

To avoid the sensitivity of sample selection to the period, based on the research of Cui (2018) [54], a comprehensive test was carried out in different periods, and the time interval was divided into three periods: 2009–2019, 2009–2014 and 2015–2019. In the first place, during 2009–2014, the regression coefficient β-value of cultivated land use efficiency in China was negative and passed the significance test at 1%, which indicates that there was absolute β-convergence. Next, the regression coefficient β of cultivated land use efficiency in China from 2015 to 2019 was negative. Nonetheless, it did not pass the significance test, which cannot explain the absolute β-convergence. Lastly, from the overall regression results from 2009 to 2019, the regression coefficient β-value was significant and negative, which proves that there was absolute β-convergence. Detailed information can be seen in Table 5.

#### 3.4.3. Conditional β-Convergence Test 

The conditional β-convergence test was used to prove whether the development trend of cultivated land use efficiency in each province will tend toward an equilibrium state and finally converge to a steady level. The conditional convergence test of cultivated land use efficiency in China was carried out using the bidirectional fixed effect model of panel data. The conditional β-convergence model of cultivated land use efficiency is designed using Wang’s research ideas for reference [55], as shown in Formula (8).
(8)$$\ln\left(Y_{i,t+1}\right)-\ln\left(Y_{i,t}\right)=\alpha +\beta \ln\left(Y_{i,t}\right)+\varepsilon_{i,t}$$

In Formula (8), $Y_{i,t}$ represents the cultivated land use efficiency of province *i* in the year *t*; *α* represents the constant term; *ε_i,t_* represents the random error term. Under individual and time two-way-fixed-effect control, the conditional β-convergence of cultivated land use efficiency in China in each period passed the significance test. Moreover, the β-values were all negative, indicating the existence of conditional β-convergence. Details are shown in Table 6.

## 4. Discussion

### 4.1. Analysis of Overall Characteristics of Cultivated Land Use Efficiency in China

Generally speaking, from 2009 to 2019, China’s cultivated land use efficiency showed a steady improvement trend, consistent with the research results of Xie, Ye and Wang [43,56,57]. Therefore, in this decade, the average utilization efficiency of cultivated land in China was between 0.85 and 0.95. Among them, the highest efficiency value in 2019 was 0.948, which is 0.028 higher than in 2009. As a result, it displays a slow upward trend, and it also shows that the provinces with low cultivated land use efficiency tend to move closer to the production frontier. In other words, the green utilization efficiency of cultivated land in the provinces with lower utilization efficiency is gradually approaching the level of those with higher green utilization efficiency of cultivated land.

In addition, it should be emphasized that from 2017 to 2019, China’s cultivated land use efficiency showed an apparent upward trend, with average use efficiency values of 0.915, 0.936 and 0.948, respectively. This is primarily because of the Action Plan for Zero Growth of Fertilizer and Pesticide Use by 2020, which was put into place in 2015 [58]. The Chinese government vigorously promotes the application of advanced agricultural technologies, such as conservation tillage, soil testing and formula fertilization and water and fertilizer integration [4,59]. As noted above, these measures can reduce the use of chemical fertilizers and pesticides, further promoting the green utilization efficiency of cultivated land.

### 4.2. Analysis of the Characteristics of Cultivated Land Use Efficiency between Provinces

From 2009 to 2019, there was a large gap in cultivated land use efficiency among provinces in China. In terms of the data, the average utilization efficiency of cultivated land in 11 provinces is greater than 1, namely in Beijing (1.267), Shanxi (1.265), Shanghai (1.131), Tibet (1.123), Xinjiang (1.105), Guangdong (1.096), Qinghai (1.063), Zhejiang (1.037), Chongqing (1.032), Jiangsu (1.026) and Henan (1.026). Thus, these data show that these provinces are located at the front of production. At the same time, the results can also indicate that, compared with other provinces, these provinces have a higher resource utilization rate and a lower environmental pollution level. On the other hand, there are four main provinces whose average cultivated land use efficiency is relatively low, and the average value is lower than 0.7. Specifically, they are Ningxia (0.676), Hunan (0.666), Jiangxi (0.635) and Gansu (0.617). In general, among the 13 major grain-producing areas in China, only Jiangsu and Henan are at the forefront of production. At the same time, Hunan and Jiangxi have low cultivated land use efficiency and have significant room for improvement.

### 4.3. Analysis of Overall Improvement Potential of Cultivated Land Use Efficiency in China

From the measurement results of this paper, China’s overall cultivated land use efficiency still needs to be improved. For the period from 2009 to 2019, the indicators with a large average slack improvement ratio are carbon emissions, pesticides and agricultural machinery. Among them, carbon emissions have the highest value, and the output ratio that can be reduced is 15.21% to reach the production frontier. Furthermore, pesticides and agricultural machinery can be reduced by 14.64% and 12.78%, respectively. By contrast, the utilization efficiency of irrigation input is the highest in cultivated land utilization, and the average reduction ratio from 2009 to 2019 is 3.55%. Moreover, the output value as a positive output shows the most significant increase in all input–output indicators. When it reaches an effective level, the corresponding improvement ratio is only slightly higher than that of irrigation, which is 3.76%.

From the change trend in time, although there are differences in the degree of change in the proportion of relaxation improvement of different input indicators, they all show a downward trend. Firstly, the reduction degree of the labor and fertilizer indicators shows a gradual downward trend with fluctuations. Secondly, the reduction ratio of pesticides fluctuates widely, showing a trend of increasing first and then decreasing. This is mainly due to the significant impact of pesticide input from 2011 to 2016, which makes the reduction range relatively large. However, it has been affected by the national fertilizer and pesticide reduction policy [58]. The gradual decline in pesticide input from 2017 to 2019 also drives the apparent reduction in the pesticide relaxation improvement ratio, thus further demonstrating that the reduction ratio of pesticides decreased to 7.89% in 2019. However, it is still higher than the reductive ratio of labor and fertilizer. Next, although the overall reduction ratio of agricultural film fluctuates greatly, it shows a downward trend. For example, in 2009, the reduction rate of agricultural film was 5.61%. The reduction rate in 2019 was 2.74%, which was the lowest for the entire period of 2009–2019. In addition, the improvement rate of agricultural film relaxation in 2019 was also the smallest.

The reduction ratio of agricultural machinery has a significant downward trend. However, the overall reduction ratio is still high, as the reduction ratio of agricultural machinery was 11.33% in 2009 and 8.48% in 2019, which are the indicators with the most significant reduction ratios among the input indicators for that year. Based on the information, the reason may be that the Chinese government began promulgating the agricultural machinery purchase subsidy policy in 2004. This further led to a continuous increase in the amount of agricultural machinery and reduced the efficiency of agricultural machinery to a certain extent. Finally, the reduction ratio of irrigation was relatively small, and showed a gradual downward trend from 4.07% of irrigation reduction in 2009 to 2.76% of irrigation reduction in 2019. Among them, the reduction ratio was the lowest in 2012, and only 1.28% of the irrigation amount was not fully utilized.

In the output index, the incremental proportion of output value is constantly increasing. However, the reduction ratio of carbon emissions has been significantly reduced. In particular, the incremental ratio of output value increased from 2.24% in 2009 to 5.28% in 2019. This shows that although the current output value is increasing rapidly, it still does not reach the frontier of production, and there is still room for improvement in the output value level. It is worth noting that among the carbon emission factors, although the carbon emissions from agricultural machinery are relatively low, they consistently showed an increasing trend from 2009 to 2019, and it is imperative to vigorously develop green agricultural machinery. The carbon emissions of fertilizers, pesticides and agricultural films account for a large proportion, but they have shown a downward trend since 2017, with room for further decline.

### 4.4. Potential Analysis of Improvement in Cultivated Land Use Efficiency among Provinces

There are apparent differences in the average relaxation and improvement ratio of input–output indicators of cultivated land use in different provinces. First of all, in light of the reduction ratio of the labor force, the highest ratio in Gansu and Shanxi is 51.06% and 42.74%, respectively, indicating that around half of the labor force input in Gansu and Shanxi is invalid. Nevertheless, Tianjin, Inner Mongolia, Hainan and Jilin have the lowest reduction ratio of the labor force, which is 0. Secondly, considering the proportion of fertilizer that can be improved, Jilin (50.53%) has the highest, while Inner Mongolia has 40.50% and Ningxia has 36.29%. The causes are excessive fertilizer input and the lack of scientific and rational fertilization in cultivated land utilization. This will consequently result in poor fertilizer utilization and possibly even soil hardening. Thirdly, Gansu has the highest ratio of pesticides (73.08%) and agricultural film (70.85%), reflecting the improved ratio of pesticides and agricultural film. Among them, the proportion of pesticide input improvement in cultivated land is much higher than in other areas, which is close to three quarters of the input amount. Moreover, judging from the proportion of agricultural machinery that can be improved, Ningxia (40.35%), Tianjin (39.39%), Anhui (36.39%), Hebei (36.19%) and other places have a large proportion of improvement. Next, from the perspective of the irrigation improvement ratio, only Ningxia (39.18%) has a considerable improvement ratio, and the improvement ratio in other areas is less than 20%. Meanwhile, Anhui (32.54%), Yunnan (31.63%) and Hunan (24.74%) are higher in terms of the proportion of output value that can be improved. Finally, from the perspective of the carbon emission improvement ratio, Jiangxi (58.05%) has the most extensive improvement ratio, followed by Hunan (42.91%) and Ningxia (28.16%). It can be concluded that the carbon emission level produced in the process of cultivated land use in these provinces is higher, thus reducing the utilization efficiency of cultivated land. In summary, as the improvement ratio of the input–output index varies between provinces, corresponding policies should be formulated according to local conditions. On the one hand, this can reduce the inefficient waste of input resources and environmental pollution. On the other hand, it will boost social and economic productivity, as well as the effectiveness of land utilization for agriculture.

## 5. Conclusions

In this paper, the concept of green development is introduced into the study of cultivated land use efficiency. Based on constructing an evaluation index system including carbon emission, which is an unexpected output, we used the MaxDEA Ultra 9.1 software to measure the cultivated land use efficiency of China’s overall and provincial regions from 2009 to 2019 by using the super efficiency MinDS model. Furthermore, the convergence of regional differences is analyzed. The evaluation index system established and the method adopted in this study are universal, and they can provide a reference for evaluating the cultivated land use efficiency in other countries. This study draws the following conclusions.

The overall utilization efficiency of cultivated land in China tended to increase slowly, from 0.920 in 2009 to 0.948 in 2019. Moreover, there are noticeable differences in cultivated land use efficiency among provinces. For example, Beijing has the highest utilization efficiency, and Gansu has the lowest utilization efficiency. There are 11 provinces where the average cultivated land use efficiency exceeds 1. Generally, in the input–output index, the carbon emission index has the largest average improvement ratio, while the irrigation index has the smallest average improvement ratio, followed by the pesticide and agricultural machinery indexes. On the other hand, there are obvious differences in the average relaxation improvement ratio of input–output indicators of cultivated land use in different provinces. Furthermore, the cultivated land use efficiency in China has absolute σ-convergence, absolute β-convergence and conditional β-convergence, which means that the difference in cultivated land use efficiency among provinces tends to decrease and China’s cultivated land use efficiency will stabilize.

Based on the above analysis conclusions, the following policy implications can be obtained.

First, it is necessary to accelerate the development of facility agriculture. Appropriate light and heat conditions can effectively promote crop growth and development and significantly positively affect the efficiency of cultivated land use. Especially in the northern area of China, where the light and heat conditions are poor in winter, there is a need to build and develop facility agriculture, such as solar greenhouses, plastic greenhouses and multi-span greenhouses. In fact, this can provide an environment for crop growth and prolong the crops’ growth period, so as to improve the annual output and income of the crops per unit of cultivated land area and the utilization efficiency of cultivated land.

Second, it is necessary to reduce carbon emissions in the utilization of cultivated land. From a national perspective, the indicator with the largest proportion of average relaxation and improvement is carbon emissions in China. Thus, it is necessary to focus on reducing carbon emissions in the process of cultivated land utilization, to achieve the goals of “carbon peaking” and “carbon neutrality” proposed by China [60]. Policies such as developing green agricultural machinery; improving the utilization efficiency of pesticides, fertilizers and water resources; and promoting the recycling of agricultural film should continue to be improved.

Third, since the improvement ratio of the input–output index is different in various provinces, corresponding policies should be formulated, depending on the improvement ratio of the input–output index of cultivated land use in different regions, to reduce the ineffective input and unexpected output as well as to improve the effectual expected output and cultivated land use efficiency.

## Figures and Tables

**Figure 1 ijerph-20-00583-f001:**
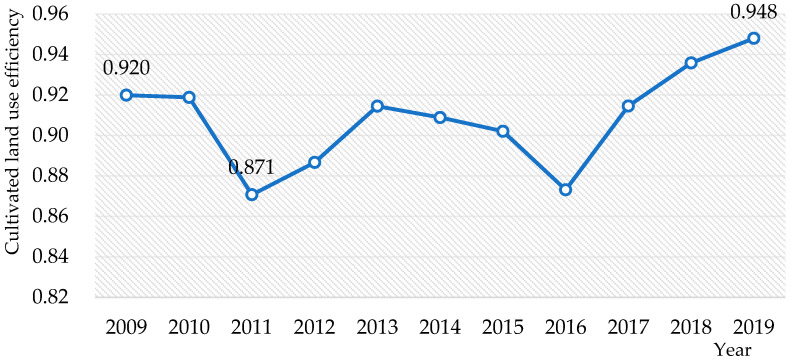
Changes in the average utilization efficiency of cultivated land in China.

**Figure 2 ijerph-20-00583-f002:**
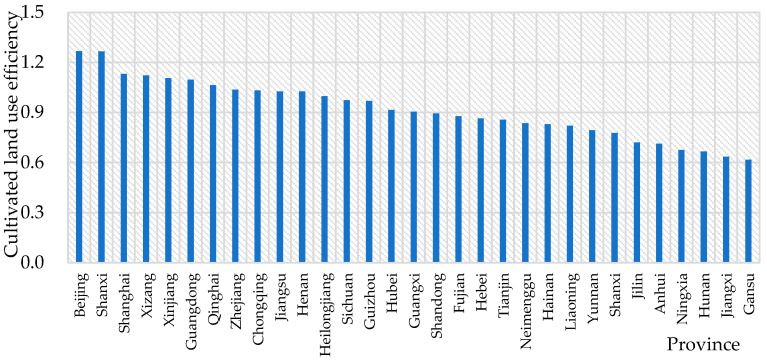
Cultivated land use efficiency in each province.

**Figure 3 ijerph-20-00583-f003:**
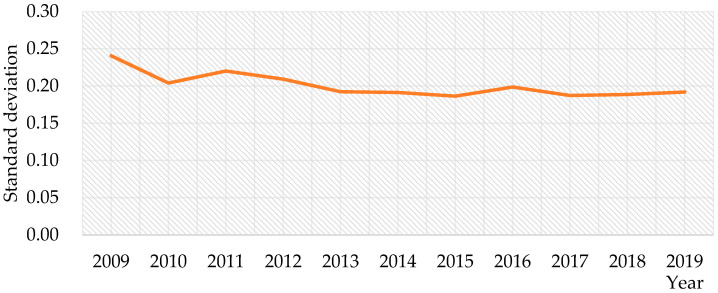
Standard deviation of cultivated land use efficiency in China.

**Table 1 ijerph-20-00583-t001:** Index system of the cultivated land use efficiency evaluation.

Indicator	Variable	Variable Description	Unit
Input	Labor input	Number of employees in the primary industry × (output value of planting industry/total output value of agriculture, forestry, animal husbandry and fishery)/total cultivated land area	people/ha
	Pesticide input	Pesticide use/total cultivated land area	kg/ha
	Fertilizer input	Usage of agricultural chemical fertilizer (pure)/total cultivated land area	kg/ha
	Power input of agricultural machinery	Total power of agricultural machinery/total cultivated land area	kw/ha
	Irrigation input	Agricultural water consumption × (output value of planting industry/total output value of agriculture, forestry, animal husbandry and fishery)/total cultivated land area	cubic meters/ha
	Agricultural plastic film input	Agricultural plastic film usage/total cultivated land area	kg/ha
Expected output	Planting output value	The gross output value of planting industry/total cultivated land area	yuan/ha
Unexpected output	Carbon emission	Total carbon emissions of cultivated land/total cultivated land area	kg/ha

**Table 2 ijerph-20-00583-t002:** Emission coefficients of major carbon sources.

Carbon Source	Carbon Emission Coefficient	Unit	Sources
Pesticide	4.9341	kg/kg	[44]
Fertilizer	0.8956	kg/kg	[44]
Agricultural film	5.18	kg/kg	[47]
Agricultural machinery	0.18	kg/kw	[44]
Irrigation	20.476	kg/hm^2^	[48,49]
Plowing	312.6	kg/km^2^	[33,50]
Rice planting	3.136	g/(m^2^×day)	[33,51]

**Table 3 ijerph-20-00583-t003:** Improvement ratio of input–output indicators of cultivated land use in China from 2009 to 2019.

Year	Labor (%)	Fertilizer (%)	Pesticide (%)	Agricultural Plastic Film (%)	Agricultural Machinery Power (%)	Irrigation (%)	Planting Output Value (%)	Carbon Emission (%)
2009	−6.59	−7.23	−5.84	−5.61	−11.33	−4.07	2.24	−18.49
2010	−8.03	−6.86	−4.36	−3.32	−11.43	−2.46	2.94	−16.83
2011	−7.84	−7.99	−21.75	−6.43	−15.21	−4.84	3.91	−20.58
2012	−7.99	−8.11	−21.05	−7.49	−13.68	−1.28	3.02	−21.25
2013	−5.72	−6.59	−18.39	−6.60	−11.68	−2.35	3.15	−16.40
2014	−3.35	−8.40	−18.35	−10.02	−13.32	−3.12	3.66	−16.23
2015	−3.65	−8.89	−20.50	−9.61	−13.18	−4.40	3.84	−13.66
2016	−8.35	−9.66	−18.86	−12.86	−19.73	−7.04	4.35	−13.54
2017	−10.12	−5.85	−14.86	−8.72	−11.21	−3.36	4.11	−10.44
2018	−8.50	−6.43	−9.24	−4.93	−11.30	−3.35	4.89	−10.31
2019	−6.04	−6.60	−7.89	−2.74	−8.48	−2.76	5.28	−9.53
Average value	−6.92	−7.51	−14.64	−7.12	−12.78	−3.55	3.76	−15.21

**Table 4 ijerph-20-00583-t004:** Average relaxation improvement ratio of input–output indicators of cultivated land use in provinces.

Province	Labor (%)	Fertilizer (%)	Pesticide (%)	Agricultural Plastic Film (%)	Agricultural Machinery Power (%)	Irrigation (%)	Planting Output Value (%)	Carbon Emission (%)
Heilongjiang	−1.42	0.00	0.00	−2.07	−5.87	−6.55	1.46	−10.00
Sichuang	−1.93	−0.50	0.00	−6.31	−0.56	−3.07	0.00	−4.21
Guizhou	−18.87	0.00	0.00	−9.88	−5.08	0.00	1.45	−14.45
Hubei	−3.97	−6.38	−14.60	0.00	−7.11	0.00	0.00	−13.31
Guangxi	−15.83	−10.71	0.00	0.00	−12.34	−1.42	0.39	−9.19
Shandong	−0.53	−3.46	−15.96	−10.44	−23.78	−2.35	5.32	0.00
Fujian	−0.09	−9.02	−16.49	−5.49	−2.64	0.00	0.00	−18.32
Hebei	−3.09	−4.74	−29.58	0.00	−36.19	−10.03	0.25	0.00
Tianjin	0.00	−27.08	−0.02	−10.83	−39.39	−4.57	1.23	−2.42
Neimenggu	0.00	−40.50	−8.13	−5.85	−32.88	−13.49	0.00	0.00
Liaoning	−0.21	−8.82	−21.96	−30.26	−4.80	0.00	1.64	−20.15
Hainan	0.00	−13.50	−42.52	−4.86	−0.84	−0.01	2.07	−17.43
Yunan	−22.85	−11.12	−7.84	−21.58	−1.73	−16.72	31.63	0.00
Shanxi	−42.74	−31.59	−39.52	0.00	−12.67	−7.36	1.03	0.00
Jilin	0.00	−50.53	−4.19	−6.16	−24.91	−1.05	21.60	−17.41
Anhui	−1.56	−12.99	−7.07	−16.83	−36.39	0.00	32.54	−13.50
Ningxia	−6.95	−36.29	−0.06	−0.68	−40.35	−39.18	7.37	−28.16
Hunan	−18.90	0.00	−16.08	−0.91	−30.11	0.00	24.74	−42.91
Jiangxi	−2.46	−3.58	−37.20	−8.39	−17.52	−1.55	21.55	−58.05
Gansu	−51.06	−4.66	−73.08	−70.85	−6.34	−8.55	11.45	0.00

**Table 5 ijerph-20-00583-t005:** Absolute β-convergence test results of cultivated land use efficiency in China.

Time Span	β-Value	*p*-Value	Adjusted R^2^
2009–2014	−0.1097	0.000	0.4000
2015–2019	−0.0275	0.228	0.0496
2009–2019	−0.0630	0.000	0.4317

**Table 6 ijerph-20-00583-t006:** Conditional β-convergence test results of cultivated land use efficiency in China.

Period	β-Value	*p*-Value	Adjusted R^2^
2009–2014	−0.3846	0.001	0.6178
2015–2019	−0.4129	0.000	0.6561
2009–2019	−0.3650	0.010	0.5226

## Data Availability

Not applicable.

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
