# Peer review of "Measuring the Cultivated Land Use Efficiency in China: A Super Efficiency MinDS Model Approach"

_ijerph, 2022, doi:10.3390/ijerph20010583_

Round 1
Reviewer 1 Report
The author reported on the cultivated land use efficiency using the data-driven modeling method. The concept is interesting and cool. The manuscript could be suitable for publication after the following reviews:
-Line 109-111, Please summarise the literature more critically and check the correct citation format. It is recommended that the conclusions corresponding to the different literature be clearly identified.
-Line 157, Add a new reference for ‘China Statistical Yearbook’. Please also pay attention to relevant issues in other parts of the paper.
-Line 241, Check the grammar ‘As shown in Figure 3’, which should be combined with the previous sentence
Before submitting again, please read the whole paper carefully and avoid punctuation, spelling, unit, and other minor problems.
Reviewer 2 Report
The authors are working on an interesting topic and quite relevant. The present a statistical approach to compute land efficiency index of harvesting rice in China at the administrative level reported by Agricultural statistical year books. The paper takes, however, a very narrow focus on the study at hand.
Some issues
Page 3 line 122. I don't think the research aims to improve cultivated land use efficiency. It rather wants to develop a metric to measure this variable. The objective needs better definition.
What is the level of spatial granularity or detail that we are aiming to measure.
Page 4 line 156. Delete is not the best choice of word here
Presentation of mathematical equations should improve
Is the expected output variable (table 1) subject to fluctuation of market prices or economic fluctuation? IT is measured as gross value. Wouldn't it be better to measure this as total production? e.g tons rice /ha?
Table 3. What is the baseline or reference value for the reported changes?
How scalable is this? can it be extended to other crops in China? How about other geographies outside of China?
The discussions and conclusions about CO2 emissions factors are not clear. It seems that some recommendations are even in contradiction with low emission development.
